Ant-mediated seed dispersal in a warmed world

Stuble Katharine L. 1 klstuble@gmail.com
Patterson Courtney M. 1
Rodriguez-Cabal Mariano A. 2
Ribbons Relena R. 1
Dunn Robert R. 3
Sanders Nathan J. 1
1 Department of Ecology & Evolutionary Biology, University of Tennessee , Knoxville, TN , USA
2 INBIOMA, CONICET, Universidad Nacional del Comahue , Bariloche, Río Negro , Argentina
3 Department of Biological Sciences, North Carolina State University , Raleigh, NC , USA
Andrew Nigel
Electronic publication date: 2014 Mar 11
Publication date: 2014
Volume: 2
Electronic Location ID: e286
Received 2013 Dec 20; Accepted 2014 Feb 2
Copyright: © 2014 Stuble et al.
Copyright year: 2014
Copyright holder: Stuble et al.
License: This is an open access article distributed under the terms of the Creative Commons Attribution License, which permits unrestricted use, distribution, and reproduction in any medium, provided the original author and source are credited.
License URL: https://creativecommons.org/licenses/by/3.0/

Keywords: Ants, Climate change, Myrmecochory, Seed dispersal, Warming

Funding: US Department of Energy PER DE-FG02-08ER64510 US National Science Foundation NSF 1136703 DOE Climate Science Center Award NSF Career Award NSF 09533390 Department of Ecology and Evolutionary Biology at the University of Tennessee Funding was provided by the US Department of Energy PER (DE-FG02-08ER64510) and US National Science Foundation (NSF 1136703) to N Sanders and R Dunn, and DOE Climate Science Center Award and an NSF Career Award to R Dunn (NSF 09533390). K Stuble was supported by an EPA STAR and funds from the Department of Ecology and Evolutionary Biology at the University of Tennessee. The funders had no role in study design, data collection and analysis, decision to publish, or preparation of the manuscript.

==============================
Climate change affects communities both directly and indirectly via changes in interspecific interactions. One such interaction that may be altered under climate change is the ant-plant seed dispersal mutualism common in deciduous forests of eastern North America. As climatic warming alters the abundance and activity levels of ants, the potential exists for shifts in rates of ant-mediated seed dispersal. We used an experimental temperature manipulation at two sites in the eastern US (Harvard Forest in Massachusetts and Duke Forest in North Carolina) to examine the potential impacts of climatic warming on overall rates of seed dispersal (using Asarum canadense seeds) as well as species-specific rates of seed dispersal at the Duke Forest site. We also examined the relationship between ant critical thermal maxima (CTmax) and the mean seed removal temperature for each ant species. We found that seed removal rates did not change as a result of experimental warming at either study site, nor were there any changes in species-specific rates of seed dispersal. There was, however, a positive relationship between CTmax and mean seed removal temperature, whereby species with higher CTmax removed more seeds at hotter temperatures. The temperature at which seeds were removed was influenced by experimental warming as well as diurnal and day-to-day fluctuations in temperature. Taken together, our results suggest that while temperature may play a role in regulating seed removal by ants, ant plant seed-dispersal mutualisms may be more robust to climate change than currently assumed.

Introduction

Understanding how organisms will respond to ongoing changes in climate, leading to subsequent changes in key ecological processes, is essential in order to predict the structure and function of ecosystems in the future (Andrew et al., 2013). For example, the alteration of interspecific interactions is one important mechanism by which climate change may ultimately alter the structure and function of ecosystems (Tylianakis et al., 2008; van der Putten, Macel & Visser, 2010; Walther, 2010; Harley, 2011; Urban, Tewksbury & Sheldon, 2012). The majority of studies on the effects of climate change on interspecific interactions have focused on negative interactions, such as competition (Suttle, Thomsen & Power, 2007), predator–prey interactions (Rothley & Dutton, 2006; Barton & Schmitz, 2009; Harley, 2011), and herbivory (Bale et al., 2002). Indeed, climate change can alter the nature and outcomes of interspecific interactions through a variety of mechanisms such as altered abundance and fitness levels of key species (Suttle, Thomsen & Power, 2007), shifts in phenology (Both et al., 2009), and species range shifts (Harley, 2011). Each of these mechanisms can disrupt interspecific interactions by altering the frequency and intensity of interactions among species.

As with most studies in ecology, work on the effects of climate change on positive interactions is lacking, even though mutualisms play fundamental roles in structuring communities and ecosystems (Callaway, 1995; Stachowicz, 2001). Mutualisms including plant–pollinator interactions and mycorrhizal interactions have been altered by climate change (Parrent, Morris & Vilgalys, 2006; Memmott et al., 2007; Hoover et al., 2012). However, the influence of climate change on other types of positive interactions is not as well studied.

One such mutualism that may be altered by climate change is myrmecochory, the ant-plant seed dispersal mutualism. This mutualism includes hundreds of ant species and thousands of plant species around the world (Beattie & Hughes, 2002; Gove, Majer & Dunn, 2007; Rico-Gray & Oliveira, 2007; Lengyel et al., 2010) and can play an important role in shaping plant communities (Bond & Slingsby, 1984). Myrmecochorous plant species typically bear seeds containing a lipid-rich fleshy appendage known as an elaiosome, to which ants are attracted. In deciduous forests of North America approximately thirty percent of understory herb species might be ant-dispersed (Beattie & Culver, 1981), and a proposed keystone seed-dispersing ant species, Aphaenogaster rudis, is responsible for upwards of 90% of ant-mediated seed dispersal (Zelikova, Dunn & Sanders, 2008; Ness, Morin & Giladi, 2009). Such specialization in interactions can make interaction networks more vulnerable to disruption as a result of low levels of functional redundancy within a system (Aizen, Sabatino & Tylianakis, 2012). Myrmecochorous plant species that rely on a single ant species (or species complex) for seed dispersal may be at increased risk for disruption by ongoing climatic change (Pelini et al., 2011a; Warren, Bahn & Bradford, 2011; Warren & Bradford, 2013) if that ant species is negatively affected by warming. Inversely, systems in which multiple species are responsible for removing seeds may prove to be more resistant to disruptions because of functional redundancy in the system (Peterson, Allen & Holling, 1998). However, despite the importance and ubiquity of myrmecochory in ecosystems around the world and the importance of temperature in regulating ant foraging, experiments examining the consequences of climatic change on this mutualism are rare (but see Pelini et al., 2011a; Warren & Bradford, 2013).

To examine the potential for climate change to alter myrmecochory, we experimentally manipulated temperature at two sites in the eastern United States (Duke Forest in North Carolina and Harvard Forest in Massachusetts) and examined overall rates of seed removal as a function of temperature. At the North Carolina site, we also examined species-specific rates of seed dispersal. We test two predictions:

Prediction 1: Rates of seed removal by ants would decrease as a result of experimental warming at the southern site where species, including A. rudis, are closer to their critical thermal maxima (CTmax) (Deutsch et al., 2008; Diamond et al., 2012a; Diamond et al., 2012b; Huey et al., 2012), a pattern previously documented in this study system (Diamond et al., 2012a). Rates of seed removal can be expected to increase at the northern site, where most species, including A. rudis, are well below their CTmax (Diamond et al., 2012a) and perhaps even below their thermal optima.

Prediction 2: Species with higher CTmax would remove seeds more readily under warmer conditions as compared to species with lower CTmax as these species are more tolerant of higher temperatures and have been found to be more active as temperatures increase (Stuble et al., 2013a).

Methods

Site description

This experiment was conducted at Duke Forest (southern site) in North Carolina and Harvard Forest (northern site) in Massachusetts, United States (U.S.), in order to examine the potential impacts of climate change on seed dispersal mutualisms near the southern and northern extents of eastern deciduous forests. The Duke Forest site consists of a closed-canopy oak-hickory (Quercus spp., Carya spp.) forest with a mean annual temperature of 15.5°C and approximately 1140 mm of precipitation annually. The Harvard Forest site is in a closed-canopy oak-maple (Quercus spp.-Acer spp.) forest with a mean annual temperature of 7.1°C and 1066 mm of precipitation a year. Though not present in the immediate vicinity of this experiment, several myrmecochorous plant species are common in these forests. These species include Asarum canadense, Asarum arifolium, Trillium spp., Viola rotundifolia, and Sanguinaria canadensis, among others. Seeds of these plant species, and myrmecochorous species in general, are typically dispersed in the spring (Thompson, 1981).

Approximately 30 ant species co-occur at the two sites, with the North Carolina site near the southern range edge and the Massachusetts site near the northern range edge for many of these species (Pelini et al., 2011b). The most abundant ant species at both sites, as is the case throughout eastern deciduous forests (King, Warren & Bradford, 2013), is the proposed keystone seed disperser A. rudis (Ness, Morin & Giladi, 2009) (or at least a species in the taxonomically vexing A. rudis complex). For the purposes of this study we are combining A. rudis, A. picea, and A. carolinensis into the A. rudis complex due to the difficultly of identifying these closely related species in the field. Crematogaster lineolata, a behaviorally aggressive species, can be abundant at the southern site in warmer months while Formica subsericea and Camponotus pennsylvanicus (also behaviorally aggressive) are the two next most abundant ant species at the Harvard Forest site (Stuble et al., 2013b).

At each site, there are twelve experimental open-top warming chambers (Fig. 1). Each chamber is 5 m in diameter and 1.2 m tall with a 2–3 cm gap at the bottom to allow ants and other organisms to move in and out. The chambers are large relative to the size of an ant (i.e., about 1000 body lengths across). Nine chambers at each site are warmed from 1.5°C to 5.5°C in 0.5°C steps using air warmed by hydronic radiators, while the three control chambers blow air at ambient temperatures into the plots (see Pelini et al., 2011b for a detailed description of the chambers). Warming treatments have been maintained continuously since January of 2010 and have been successful at maintaining the targeted temperature increases. For 2011, a significantly positive relationship between the target temperature and actual temperature increase was maintained (p < 0.01, R2 = 0.99).

Figure 1 Warming chamber at Duke Forest.

The chambers contain about one A. rudis colony per square meter while the average foraging distance of an A. rudis colony is ∼70 cm (L Nichols, unpublished data, 2012). In addition, during the summer of 2012, we watched 72 A. rudis workers visiting baits and returning to their colonies. Out of those 72 observations, only 1 worker visited a bait in the chamber and returned to a nest outside the chamber (L Nichols, unpublished data, 2012). So, most of the activity we see is from ants in the chambers. Significant shifts in levels of foraging activity at food baits have been documented in the chambers across the temperature treatments for a variety of ant species, with more thermally tolerant species exhibiting higher levels of foraging activity in warmer chambers than species with lower thermal tolerances (Stuble et al., 2013a). Those results suggest that temperature does mediate foraging behavior. Such a result, that environmental context can mediate foraging behavior, is in line with previous work in this system (Pelini et al., 2011a) and others (Cerdá, Retana & Cros, 1997; Sanders & Gordon, 2000; Sanders & Gordon, 2003; Gibb & Parr, 2010).

Seed removal

To assess the impact of temperature on rates of seed dispersal, we haphazardly positioned one seed cache in each of the 12 chambers at Duke Forest and Harvard Forest. Each cache contained 20 seeds of the myrmecochorous species Asarum canadense placed on a laminated index card. The range of Asarum canadense extends from New Brunswick, Canada to North Carolina in the southern US (Cain & Damman, 1997). Seeds of A. canadense are similar in mass to many other myrmecochorous plant species (Michaels et al., 1988), including the locally common Sanguinaria canadensis and Asarum arifolium, and are readily removed by A. rudis (Turner & Frederickson, 2013). Seeds used in the trials at Duke Forest were collected at North Carolina State University’s Schenck Forest in Raleigh, North Carolina on May 11, 2011 and those used in the Harvard Forest trials were collected from Mt. Toby in Massachusetts on June 8, 2011 when seeds of this species naturally dehisce at these locations. Seeds were kept frozen until used in a trial (Morales & Heithaus, 1998). We covered each seed cache with a mesh cage (14.25 cm long × 14.25 cm wide × 7.5 cm tall, mesh size 1 cm × 1.5 cm) to allow ants to access the seeds while preventing access by rodents. Caches were left out for one hour, after which time the number of seeds remaining in the cache was counted and any remaining seeds were removed from the chamber. Though observing seed removal for an hour limits our ability to account for the fate of all seeds, using this standard timeframe allowed us to compare relative rates of seed removal across treatments. A total of ten trials (one seed cache deployed per chamber) were conducted at Duke Forest between May 12 and May 25, 2011, with five trials conducted during the day (between 0900 and 1900) and five during the night (between 2100 and 0500). Another five trials were conducted at Harvard Forest between June 16 and June 30, 2011: three during the day and two at night. These dates corresponded with the time periods during which the seeds were naturally released at each site, as opposed to conducting this experiment in the hottest part of the year when the impacts of warming might be expected to be greater, but when any results might be less ecologically relevant.

We calculated the average seed dispersal rate (number of seeds removed in an hour) for each chamber at each site. We used ANCOVA to examine differences in seed dispersal rates as a function of temperature treatment (which we refer to as Δ°C, included as a continuous variable) and site. The number of seeds removed per hour was square root transformed to meet assumptions of normality. All statistics were performed in SAS, version 9.2.

To determine the ant species responsible for removing the seeds, we continuously observed caches of 10 A. canadense seeds within the chambers at Duke Forest for one hour, or until all seeds were removed. Four seed removal observations were conducted in each chamber: two during the day and two during the night. Nighttime observations were conducted using red lights, which is typical in studies of ant behavior at night (Hodgson, 1955; Narendra, Reid & Hemmi, 2010). We recorded the identity of the ant species removing the seeds. When possible, we also followed the seed back to the nest (or under leaf litter in some cases) and noted the distance it had been moved. At the beginning of each observation, we took four ground surface temperature measurements using a handheld infrared thermometer (Raytek® Raynger ST, + /−1°C), one at each corner of the seed cache, which were averaged together. These temperature readings provided us with estimates of ground-surface temperature conditions in the immediate vicinity of the seeds. Ground-surface temperature has been shown to be an important driver of foraging activity in ants (Whitford & Ettershank, 1975; Crist & MacMahon, 1991). We calculated the percentage of seeds removed by each species overall, as well as separately for day and night. We also calculated the mean number of seeds removed by each species in each chamber across all trials.

We used linear regressions to examine the relationship between seed dispersal rate and temperature treatment for each ant species. (We examined several polynomial regressions, but found none of them to be a better fit than simple linear regressions. Generalized linear models also yielded qualitatively similar results.) Mean numbers of seeds removed (per species and chamber) were log transformed to meet assumptions of normality for A. rudis and C. lineolata.

Finally, we calculated the average ground surface temperature (based on temperatures collected with the infrared thermometer) at which each species removed seeds across all treatments and times. We then examined the relationship between the average temperature at which a species removed seeds and the CTmax of that species (as calculated by Diamond et al., 2012b at or near the study site at the same time of year as this study was conducted) across all species observed removing seeds in the system. Aphaneogaster lamellidens was excluded from this analysis as it was only observed removing seeds from two seed caches and was an outlier (as indicated by a plot of residuals by predicted values).

Results and Discussion

Seed removal rate did not depend on temperature treatment (°C above ambient) and did not vary between sites (F2,21 = 0.93, p = 0.41; Fig. 2). This result is despite the fact that most of the foragers observed in this study were from colonies within the experimental chambers. At the southern site, where seed dispersal observations were conducted, the mean seed removal distance was 51 cm, and only 2% of observed seeds were removed more than 2 m. The lack of response to experimental warming contrasts with the prediction that, based on the thermal limits of A. rudis and its disproportionate role in seed dispersal, seed dispersal rate should decline with increasing temperatures. Regardless of temperature treatment or site, ants removed ∼23% of seeds per hour (an average of 4.6 seeds out of 20). At the southern site, we observed seven ant species removing seeds across a range of ground surface temperatures from 17°C to 30°C (Table 1). Aphaenogaster rudis was the most common seed disperser, removing approximately 45.5% of seeds (Table 2). However, there was no relationship between the rate of seed dispersal by A. rudis and temperature treatment (Table 1). With the exception of C. lineolata, which showed a marginally significant increase of approximately 0.1 seeds removed per degree of warming, seed removal did not vary systematically with temperature treatment for any ant species (Table 1). This finding is despite previously observed shifts in foraging under experimentally warmed conditions (Pelini et al., 2011a; Stuble et al., 2013a). Pelini et al. (2011a) found an approximately 50% decrease in several types of foraging, including seed removal, as a result of 1°C of warming at the southern site, though no change was observed at the northern site. Using the same warming chambers as in this study, Stuble et al. (2013a) found species-specific shifts in foraging activity as a result of experimental warming consistent with the thermal tolerances of the foraging species. Further, ant community composition shifts in response to experimental warming, demonstrating the importance of temperature in regulating the ant community (Diamond et al., 2012a). Despite this, experimental warming apparently does not affect the aspects of the seed-dispersal mutualisms we studied in this system. This begs the question—why isn’t this seed-dispersal mutualism disrupted by experimental warming?

Figure 2 Number of seeds removed (± standard error) in the course of an hour as a function of temperature treatment.

Black dots represent Duke Forest and gray dots represent Harvard Forest.

Table 1 Test statistics are from linear regressions examining the influence of temperature treatment on the number of seeds removed by each ant species (d.f. = 11 for all species).

Species	F	p	
Aphaenogaster lamellidens	2.58	0.58	
Aphaenogaster rudis	2.45	0.15	
Camponotus castaneus	0.10	0.75	
Camponotus pennsylvanicus	0.24	0.64	
Crematogaster lineolata	4.14	0.07	
Formica pallidefulva	0.34	0.58	
Formica subsericea	0.02	0.90	

One possible answer is that foraging behavior by Aphaenogaster rudis may be more tolerant of experimental warming than previously thought. A. rudis is a keystone mutualist in this and other systems, responsible for the majority of ant-mediated seed dispersal (Zelikova, Dunn & Sanders, 2008; Ness, Morin & Giladi, 2009; Canner et al., 2012). The abundance and activity of A. rudis declines with elevation (i.e., lower temperatures) at biogeographic scales (Zelikova, Dunn & Sanders, 2008). Additionally, the relatively low thermal tolerance of this species accurately predicts its activity relative to other species (Stuble et al., 2013a). However, when exposed to experimental warming, the abundance (Pelini et al., 2011a) and foraging activity (Stuble et al., 2013a) of A. rudis apparently do not decline at either study site. Importantly, the average foraging distance of Aphaenogaster spp. was ∼70 cm at the study site (L Nichols, unpublished data, 2012). Further, based on the proportion of these ants observed foraging into the chambers from outside in observations, we’d predict that only about one of the eighty seeds observed being removed by A. rudis was likely to have been removed by a worker originating outside of the chambers. Thus, it is not likely the case that ants are coming to baits from nests that are outside the chambers. However, even in cases in which individual workers do forage at the experimental baits from colonies outside of the chambers, these individuals are still exposed to the experimental temperature conditions while discovering, foraging at, and recruiting to the seed caches. These results suggest that temperature does not substantially alter this foraging behavior. The apparent tolerance of the foraging activity of this important seed dispersing species to warming may play a major role in promoting the stability of ant-plant seed dispersal in light of global change.

Table 2 Percentage of seeds removed by each species overall, during the day, and the night.

Species	Overall percent	Day percent	Night percent	
Aphaenogaster lamellidens	8.5	17.2	0.0	
Aphaenogaster rudis	45.5	48.3	42.7	
Camponotus castaneus	26.7	0.0	52.8	
Camponotus pennsylvanicus	2.8	2.3	3.4	
Crematogaster lineolata	6.8	12.6	1.1	
Formica pallidefulva	2.3	4.6	0.0	
Formica subsericea	7.4	14.9	0.0	

It is important to note that six ant species other than A. rudis were observed removing seeds in this study, and they removed >50% of the seeds. This runs counter to several studies suggesting seed dispersal mutualisms may be highly specialized (Gove, Majer & Dunn, 2007; Ness, Morin & Giladi, 2009). Both the foraging activity and abundances of several of these species, including C. lineolata and Formica pallidefulva, shift with warming (Pelini et al., 2011a; Diamond et al., 2012a; Stuble et al., 2013a), resulting in an altered community of foragers (Diamond et al., 2012a). By having multiple ant species interacting with myrmecochorous plants, this ant-plant seed dispersal mutualism may be relatively resistant to the effects of warming as some ant species increase in activity and abundance while others decline in abundance with temperature. Previous work on ant foraging and community composition as a result of the experimental warming at these sites suggests that species vary in their responses to warming, which might moderate the overall effects of climatic warming on entire assemblages (Stuble et al., 2013a).

In addition to the apparent (and of course relative) resistance of the foraging of A. rudis to warming, along with the diversity of ants engaging in this mutualism, another factor possibly strengthening the resistance of myrmecochory to warming may be the timing of ant-mediated seed dispersal within deciduous forests of the eastern US. Ant-dispersed seeds in these forests, including those of Asarum canadense, are primarily dispersed in the spring (Thompson, 1981). Temperatures in May in North Carolina and June in Massachusetts at the study sites are far from the critical thermal maxima of ant species in the system. For example, the critical thermal maximum for A. rudis is 38°C and 40°C for populations at the northern and southern sites, respectively (Diamond et al., 2012b), as opposed to the mean environmental temperatures during the sampling period, which were 20°C at the northern site and 22°C at the southern site. The thermal buffer between CTmax and the environmental temperature during the time of year when seeds are dispersed may confer some degree of tolerance on this mutualism. Pelini et al. (2011a) found that rates of seed removal decreased in a passive experimental warming at the same two sites, despite achieving warming of only 0.3°C above ambient. However, the seed removal trials in Pelini et al.’s experiment were conducted mostly in August when ambient environmental temperatures are hotter than those experienced in the present study. We suggest that the proximity of ants to their upper thermal limits in August may have driven the effects of warming observed in the Pelini et al. (2011a) study while seed dispersal occurring in the spring when our study was conducted may be less likely to be detrimentally impacted by warming. However, this protection assumes that the peak of A. rudis activity and seed set coincide. Phenological shifts in plant reproduction caused by ongoing warming (Price & Waser, 1998; Dahlgren, von Zeipel & Ehrlén, 2007; Inouye, 2008; Liu et al., 2011; Wolkovich et al., 2012) have the potential to result in seeds appearing before ants become active (Warren, Bahn & Bradford, 2011). Warren, Bahn & Bradford (2011) suggest that while both seed release by plants and onset of foraging in ants seem to be driven by temperature, variability in activation temperatures among ant species may result in situations in which early seeding plant species may become decoupled from their foragers in some areas.

Despite the apparent tolerance of myrmecochory to experimental warming in this study, there was a significant relationship between the ground-surface temperature at which a species removed seeds and the critical thermal maximum of that species (F1,4 = 7.35, p = 0.05, R2 = 0.65, Fig. 3). That is, those species with high thermal tolerances were most active under the warmest temperatures. The positive relationship between CTmax and seed removal temperature suggests that while chronic experimental warming may not affect rates of seed dispersal, temperature does relate to rates of seed removal. This finding incorporates both temperature variability associated with the temperature treatments as well as daily temperature variability and complements other studies that have shown physiological tolerance to be an important predictor of ant activity (Diamond et al., 2013).

Figure 3 Temperature (± standard error) at which seeds were removed as a function of a species’ critical thermal maximum (CTmax).

One important caveat to our study (and to most studies of myrmecochory) is that we do not know the ultimate fate of the seeds once the ants removed them. It is possible that warming could still alter the dynamics of plant populations by altering rates of germination and seedling survival post-germination (De Frenne et al., 2012), even in cases in which seed dispersal remains unaffected as temperatures increase. Additionally, some species, including C. lineolata, dispersed seeds very short distances (only a few centimeters) while other species, such as C. castaneus, often carried seeds several meters (Ness et al., 2004, personal observation) and species may vary in seed handling and where they ultimately discard the seed (Hughes & Westoby, 1992; Giladi, 2006; Servigne & Detrain, 2010; Stuble, Kirkman & Carroll, 2010). As such, even slight shifts in relative rates of dispersal among these species may alter plant population dynamics if dispersal distances and seed fate differ substantially among species (Bond & Slingsby, 1984). Similarly, we have grouped three species into the Aphaenogaster rudis complex. These species may differ in their thermal niches and, as such, may respond differentially to warming. Past research has, in fact, suggested that some of these species may be prone to phenological mismatches with their plant partners as a result of climatic warming (Warren, Bahn & Bradford, 2011; Warren & Bradford, 2013; Warren & Chick, 2013). Finally, by observing seed caches for an hour, we fail to collect data on the fate of seeds not removed in that time frame. However, a considerable proportion of the seeds that will be removed by ants are, in fact, removed soon after release from the parent plant (Turnbill & Culver, 1983; Beaumont, Mackay & Whalen, 2013) and seeds not dispersed by ants are at risk of predation by rodents (Heithaus, 1981). Additionally, as in most studies of ant–seed interactions, we focus on seeds of only a single plant species. Including seeds from more species would be ideal, but would have been beyond what was possible in this (or in most) studies.

It has become axiomatic that interactions among species are being affected by ongoing climatic change. However, in this study, we found no reduction in overall rates of seed removal as a result of experimental warming. Moreover, there were no differences in species-specific seed removal rates at the southern site. We suggest that myrmecochory may be resistant, at least in part, to climatic warming as a result of the diversity of ants active in this mutualism as well as the seasonal timing of this mutualism in the spring when most ant species are far from their upper thermal limits. Importantly, however, if phenological mismatches arise, or if the fate of seeds after dispersal is altered, the consequences of warming on plant populations and communities could emerge in unexpected ways.

Supplemental Information

Supplemental Information 1 Seed dispersal data

Includes data on overall number of seeds removed per hour (out of 20) and number of seeds removed per species in separate observed seed dispersal trials.

Click here for additional data file.

We thank M Burt and I Del Toro for help in the field. A Ellison and N Gotelli were integral in developing and improving this project. We also thank K Prior, R Warren, and two anonymous reviewers for comments that aided in improving this manuscript.

Additional Information and Declarations

Competing Interests

Author Contributions

NJ Sanders is an Academic Editor for PeerJ.

Katharine L. Stuble conceived and designed the experiments, performed the experiments, analyzed the data, wrote the paper, prepared figures and/or tables, reviewed drafts of the paper.

Courtney M. Patterson performed the experiments, analyzed the data, wrote the paper, reviewed drafts of the paper.

Mariano A. Rodriguez-Cabal conceived and designed the experiments, performed the experiments, wrote the paper, reviewed drafts of the paper.

Relena R. Ribbons performed the experiments, wrote the paper, reviewed drafts of the paper.

Robert R. Dunn conceived and designed the experiments, contributed reagents/materials/analysis tools, wrote the paper, reviewed drafts of the paper.

Nathan J. Sanders conceived and designed the experiments, analyzed the data, contributed reagents/materials/analysis tools, wrote the paper, reviewed drafts of the paper.

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
