# Peer review of "Ant-mediated seed dispersal in a warmed world"

_PeerJ, doi:10.7717/peerj.286_

## Round 0.1 · original submission · Minor Revisions

The reviewers have made some good comments to strengthen the manuscript and to fix some editorial issues (e.g. lack of italicising species names). I also have some comments that need to be addressed before the manuscript is acceptable for publication

Itaclise all genus/ species names

Line 41. Consider citing PeerJ paper
Andrew NR, Hill SJ, Binns M, Bahar MH, Ridley EV, Jung M-P, Fyfe C, Yates M & Khusro M (2013) Assessing insect responses to climate change: What are we testing for? Where should we be heading? PeerJ 1: e11 doi 10.7717/peerj.11.

Line 91. Are you referring to Stubble et al 2013 a or b?

Line 117. Why is it important that the chambers are 1000 ant body lengths in diameter? It is an unusual measurement unit.

Line 157. Since you seem to have replicated seed dishes to measure dispersal in each chamber, you would be better using you replicates and the data and carry out a generalized linear model (GLZ) with a binomial distribution and a logit-link function test rather than a simple chi square.

Line 185. You are basing your CTmax data on Diamond, but this value can change between sites and time of year. Were your samples taken at the same time of year as Diamonds?
Line 223. Stuble et al a or b??
Line 246. Wording ‘quite a bit of variability’ is vague. Be more specific.
Line257. What were the ground temperatures at these sites? This can be critical when measuring CTmax. See Andrew NR, Hart RA, Jung M-P, Hemmings Z & Terblanche JS (2013) Can temperate insects take the heat? A case study of the physiological and behavioural responses in a common ant, Iridomyrmex purpureus (Formicidae), with potential climate change. Journal of Insect Physiology 59: 870-880 doi 10.1016/j.jinsphys.2013.06.003.

·

Basic reporting

Stuble et al. conducted a study that examined the effect of experimental warming on ant-mediated seed dispersal. They subjected ant communities at a southern site and at a northern site to experimental warming and measured seed dispersal in a plant species that is adapted to dispersal by ants. This is well-designed and well-presented study that makes a good contribution to understanding how climate change influences positive species interactions.

They manuscript is well-written and contains appropriate background information and references to the literature. The figures and tables are appropriate and the submission includes relevant and complete results. I only have a few minor suggestions (see general comments).

Experimental design

The research question is clearly defined and the design of the experiment is sound. My only suggestion (other than a few minor comments outlined the in the general comments section) is that more detail should be added to the methods section to explain what is known about the density and distribution of ant nests in the experimental chambers.

Consider breaking up the paragraph (L115-135) into two paragraphs. The first paragraph could describe the chambers and temperature treatments. The second paragraph could describe the distribution and density of ant nests and species in the chambers. This would allow for more room to explain this aspect of the design.

For example, the following details regarding the density and distribution of ants should be clarified: L118-L119 it is not clear here if this statement is referring to A. rudis only or all ant species (I assume the former). Was the estimation of 1 nest per m2 quantified by the authors? If so, the authors should provide statement that they produced this estimate and refer to this as unpublished data. If the authors did not quantify A. rudis nest density at their sites or in the chambers, then the authors could reference other studies that report high densities of A. rudis nests in forest ecosystems (Lubertazzi 2012 Psyche, also see references in Ness et al. 2009) and then say they assume similar densities at their sites (and in chambers). Also, the authors should add in some comments about the likelihood that nests of other ant species are situated within the chambers. I am assuming that there is much more variation in where ant nests of species (other than A. rudis) are situated relative to the experimental chambers. The authors make multiple statements in the text (L122, L191) that the treatments affect ants situated in the chambers. I am assuming that they are referring to A. rudis only and not all ant species, but as the methods are currently written this detail is unclear.

Validity of the findings

Minor comments in general comments section

Additional comments

Here are a few minor suggestions:

L22 consider replacing “US” with “North America”, as this ecosystem extends into Canada; also in L64

L24 consider replacing “removal” with “dispersal”

L21 sometimes the mutualism is referred to as “ant-plant seed dispersal” and sometimes “ant plant seed-dispersal” throughout the manuscript

L62 this would be a good spot in the manuscript to define the term myrmecochory as ant-plant seed dispersal. Also consider using this term more often throughout the paper, instead of “ant-plant seed dispersal”.

L64 consider adding a line stating that myrmecochory is integral in shaping plant communities (e.g., Christian 2001, Nature)

L69 Myrmecochorous is spelled incorrectly

I think that it would be helpful to more specifically describe what a myrmecochorous plant is – i.e., plants that have seeds containing a fleshy lipid-rich appendage (elaiosome) that are attractive to ants

L70 consider replacing “single” with “a single or a few ant species”, as most systems seem to be dominated by keystone dispersers that are not a single species, but a group of species in a genera (e.g. Aphaenogaster (Ness et al. 2009), Rhytidoponera (Gove et al. 2007)).

L79 state southern and northern site here, given that these terms are used in the prediction statements

L106 consider moving this line up (i.e., as the second sentence in the methods). If not, consider removing the statement “near the southern extent and northern extent of several ant species” from L97.

Consider moving the statements describing the A. rudis complex in L112-114 up to L109 where the A. rudis complex is first mentioned. Alternatively, remove “(or at least a species in the A. rudis complex)” from L109 (as the A. rudis is not defined at this point).

L117 I don’t think that this statement is necessary

L119 consider replacing “moreover” with “in addition”; and “watched” with “observed”

L127 consider adding “For example in 2011, we found a… (unpublished data)”

L139 what were the seeds placed in or what is a cache (e.g. a petri dish, on the ground)? How big is a cache?

L142 consider adding a line stating that A. canadensis seeds are readily picked by A. rudis (e.g., Turner & Frederickson, 2013, PLOS one, and look through references in Table 2 in Ness et al. 2009)

L143-145 consider adding, “when seeds naturally dehisce in these locations”

L145 add the dimensions of the mesh cage and the mesh size

L151 were trials conducted on different days? If so, were the seeds stored in any particular way between collection and the trials given that elaiosome quality can degrade quickly once seeds are collected?

L288-L292 I think that this is an important point that could be discussed in more detail. Ant partner identity can be important in influencing the outcome of myrmecochorous interactions. Differences in body size, foraging behavior, or nest characteristics, for example, could determine the rate at which seeds are picked up, how far they are moved (as you mention), how they are processed in the nest, and where they are ultimately deposited (Giladi 2006 Oikos; e.g., Hughes & Westoby 1992 Ecology, Ness et al 2004 Ecology, Servigne & Detrain 2010 Ecological Research, Prior et al. in press Ecological Entomology). Further, some species picking up seeds (other than A. rudis) could act as low quality dispersers (e.g., damage seeds, or remove the elaiosome in situ without moving seeds) (Giladi 2006). These differences could scale up to have large effects on plant communities (Christian 2001). Thus, even though overall dispersal is not effected by temperature, alterations in which ant species move the seeds under different temperatures could potentially have large effects on plant communities.

Figure 2: error bars represent…

·

Basic reporting

Stuble et al. examine the flexibility of ant-plant seed mutualisms by observing seed removal rates and ant species in field warming chambers. Overall, I think this an interesting, well-written study and reports important results not only on ant-plant interactions but general climate change research into community interactions. I made lots of little comments and note several papers on which I am author (not for shameless self promotion, but they seem very relevant, and I am most familiar with them) – all of which are just suggestions and not requirements for publication.

Experimental design

The experimental design appears robust and relevant to the study system.

Validity of the findings

I find that the methods and statistics seem a bit shaky, or they are not reported well and need clarification. I include individual comments in "General Comments for the Author"

Additional comments

Line 45 – May want to site: (Urban et al. 2012)

Line 47 – awkward sentence beginning, “Though rare – relative to studies that … – empirical studies…” Better yet, you may want to rephrase the sentence as it is very long.

Lines 55-60 – Very relevant to this paragraph: (Warren and Bradford 2013)

Line 64 – “In deciduous forests of the U.S., approximately …”

Line 66 – italics: “Aphaenogaster rudis,” and throughout.

Line 72 – or, if as shown by Warren and Bradford (2013), key ant species shift with warming.

Line 79 – Give “United States (U.S.)” at first use and then just “U.S.” afterward.

Lines 81-82 – nitpicky, but you cannot have a posteriori predictions (predictions are a priori by nature).

Line 83 – remove “would” – unneeded modifier

Line 96 – above you use “United States” but here you use USA, which is the abbreviation for United States of America. Choose one.

Line 99 and 101 – my guess is that something stripped out your italics, because you have more species names not in italics here.

Line 102 – you have not explained “the chambers” yet so a bit more description might be necessary.

Lines 104-105 – this sentence seems tacked on here. Do you mean that seeds of the mentioned species or all myrmecochores? You may want to expand for those not familiar with myrmecochory that myrmecochores are dispersed relatively early in spring compared to other non-myrmecochorous plants.

Lines 106-107 – share? All 30 species occur at both sites?

Line 108 – also, it is probably relevant that A. rudis species are the most abundant arthropods in eastern deciduous forests (King et al. 2013)

Lines 112-114 – even though several studies suggest species-specific temperature requirements between at least some of these species (Warren et al. 2011a, Warren et al. 2011b, Warren and Chick 2013)? It seems that this issue should be addressed, at least somewhat.

Lines 115 on – note that sometimes you place the unit of measurement with the number with spacing (e.g., 5 m) and other times to omit the space (e.g., ~70cm). You want to be consistent with the spacing.

Lines 122-135 – Aphaenogaster spp. are very sensitive to humidity; are there any data on the impacts on chamber humidity by the warming?

Lines 141-142 – You may want to note that A. canadensis seeds are similar to Sanguinaria canadensis, which I am guessing occurs at both of your sites, and other Asarum spp. (formerly Hexastylis) that occur at your Duke site.

Line 146 – Iinteresting that you are looking at changes in species interactions with warming, but leave out what is possibly a key interaction (see Giladi 2006) by excluding rodents. Did you collect any data on seed predation by rodents as a function of rodents? Any observations?

Line 153 – Semi-colons separate two independent clauses; you separate an independent and dependent clause with one here.

Lines 150-151 – Is a “trial” one bait offering per chamber so that each of 10 trials occurred in different chambers?

Lines 157 – This analysis seems a bit problematic. You have 10 Duke observations and 5 Harvard observations, and use Site as a treatment. As one would expect more variance, and hence less power, with half the Harvard as Duke observations, the Site variable is suspect.

Lines 160 – Did you conduct a test to show that data were not normally distributed? Did you conduct a test afterward to test whether your transformation worked? Though often done, rote data transformation is not a good idea. Even better, your data are count data, a GLM using Poisson distributed error probably would be better.

Lines 178-181 – Linear regressions are not used to examine “differences” across treatments (which are discrete) unless in an ANOVA structure. Do you mean “across” treatments. Same comments on transformation as above.

Lines 182-188 – If you took the average temperatures, what are the replicates?

Lines 190-191 – You note above that temperature is included as a continuous variable, hence the ANCOVA with Site, but show a figure with temperature as discrete “treatments.” I think you want to say that you examined seed removal as a function of temperature. You did not conduct trials outside of chambers, which would then make the chambers a treatment, but instead looked at variation in response by ambient + chamber temperature. Hence, calling temperature a treatment seems inaccurate here unless I am missing something.

Really, what you probably (but not necessarily) should be doing here is seed removal as a function of temperature in an Generalized Linear Mixed Model with site and/or chamber as a random effect (to account for the autocorrelation) and the error as a Poisson distribution.

Lines 191, 204 – “This” should modify something. I think you are saying “This result.”


Giladi, I. 2006. Choosing benefits or partners: a review of the evidence for the evolution of myrmecochory. Oikos 112:481-492.
King, J. R., R. J. Warren, and M. A. Bradford. 2013. Social insects dominate eastern US temperate hardwood forest macroinvertebrate communities in warmer regions. PLoS ONE 8: e75843.
Urban, M. C., J. J. Tewksbury, and K. S. Sheldon. 2012. On a collision course: competition and dispersal differences create no-analogue communities and cause extinctions during climate change. Proceedings of the Royal Society B-Biological Sciences doi: 10.1098/rspb.2011.2367.
Warren, R. J., V. Bahn, and M. A. Bradford. 2011a. Temperature cues phenological synchrony in ant-mediated seed dispersal. Global Change Biology 17:2444-2454.
Warren, R. J. and M. A. Bradford. 2013. Mutualism fails when climate response differs between interacting species. Global Change Biology Early View:doi: 10.1111/gcb.12407.
Warren, R. J. and L. Chick. 2013. Upward ant distribution shift corresponds with minimum, not maximum, temperature tolerance. Global Change Biology 19:2082-2088.
Warren, R. J., P. McAfee, and V. Bahn. 2011b. Ecological differentiation among key plant mutualists from a cryptic ant guild. Insectes Sociaux 58:505-512.

---

## Round 0.2 · accepted · Accept

Dear Katharine,

You and your co-authors have done a good job in responding to the reviewers queries and questions/ concerns. Will be good to see the work fully published in PeerJ. Congrats Nigel